# Accuracy Maps of Weigh-In-Motion Systems for Direct Enforcement

Janusz Gajda , Piotr Burnos , Ryszard Sroka * and Mateusz Daniol

Department of Measurement and Electronics, AGH University of Science and Technology, Al. A. Mickiewicza 30, 30-059 Kraków, Poland; jgajda@agh.edu.pl (J.G.); burnos@agh.edu.pl (P.B.); daniol@agh.edu.pl (M.D.)
* Correspondence: ryszard.sroka@agh.edu.pl; Tel.: +48-12-617-28-00

**Abstract:** The need to protect road infrastructure and the environment, as well as to increase the safety of road users and to ensure fair conditions of competition in road transport, requires an increase in the efficiency of the elimination of overloaded vehicles from road traffic. The replacement of "manual" vehicle control (carried out by inspectors of the relevant services) by automatic control can ensure that these are highly effective. Such control can be implemented directly on the basis of weighing results obtained from weigh-in-motion (WIM) systems. The high sensitivity of WIM systems to various interfering factors is an obstacle to the full implementation of this goal. This paper presents a concept for accuracy maps determined for direct enforcement WIM systems. The use of such maps allows for the minimization of the probability of an error consisting in classifying a normative vehicle as an overloaded one.

**Keywords:** WIM; vehicle weight control; enforcement weigh-in-motion systems; accuracy of WIM systems; accuracy maps

## 1. Introduction

The need to protect road infrastructure and the environment, to ensure the safety of road users, and to ensure fair competition in road transport imposes an obligation on road managers to check the weight of vehicles. This obligation is carried out by specialized services. However, the effectiveness of this action is significantly reduced due to the use of static or low-speed weighing systems and the "manual" implementation of the vehicle weighing procedure. The development of the technique and technology of measurement tools used to control the weight of vehicles allows the use of weigh-in-motion systems in the "direct mass enforcement" mode. In many countries, work is under way to implement this idea. Currently, WIM systems function mainly as pre-selection systems, supporting the static weighing of vehicles [1].

In [2,3], general recommendations regarding the selection of the location of WIM systems, the procedures used during their installation, the procedures for calibrating such systems, and the assessment of their accuracy are presented. The main issue addressed in this document is the assessment of the accuracy of WIM systems.

Reference [4] contains guidelines for the calibration of automatic instruments for weighing road vehicles in motion, with particular emphasis on measurements performed during calibration, the method of processing measurement results, determination of uncertainty, and the content of the calibration certificate.

Reference [5] presents information on the introduction of the National Dynamic Axle Weight Measurement System in Hungary. There are over 100 checkpoints on the roads. The article presents a brief summary of the impact of high-speed WIM systems on-road use. It contains detailed statistical analyses of the preliminary results.

Reference [6] concerns the results of the WIM system research. The aim of the research was to determine the conditions and to develop weighing algorithms necessary to ensure

an appropriate level of confidence for the weighing results. A sufficiently high level of confidence is necessary for the legalization of the HS-WIM system operating in the direct mass enforcement mode.

References [7–10] are of a legislative nature. They have been developed by national metrology institutes (NMIs) in various countries. The aim is to enable metrological legalization of WIM systems operating in the direct mass enforcement mode. The large set of countries introducing such regulations indicates the great importance of the problem of continuous and effective control of the weight of motor vehicles. Weighing in motion (WIM) is defined as a process of measuring the dynamic forces exerted by the wheels of a moving vehicle and estimating the static wheel load of the weighed vehicle. They specify the metrological and technical requirements for WIM systems used to carry out weighing in motion understood in this way. Test methods for type approval and metrological verification procedures for such systems are specified.

Reference [11] describes the HS-WIM system, whose measurement capabilities have been extended by additionally equipping it with a vehicle weighing system at low speed, i.e., in the stop-and-go mode. The described system is capable of weighing both heavy, multi-axle vehicles and passenger cars with the same accuracy.

In References [12,13], the authors presented the results of research on the properties of WIM systems, which are important from the point of view of using these systems in the direct mass enforcement mode. In particular, the quantitative influence of environmental factors on weighing accuracy was determined. The possibility of using multisensor systems, i.e., WIM systems equipped with many load sensors, was also studied.

Reference [14] presents a summary of a large project launched in 2014 by the French Ministry of Ecology, Sustainable Development and Sea, in charge of transport. The aim of the project was to demonstrate the possibility of using WIM systems for direct enforcement of regulations specifying the permissible weight of motor vehicles and the maximum value of the axle load.

Work is now underway in two areas: legislative and technical. The first area concerns the establishment of a law that will enable the legal use of WIM systems for enforcement. In turn, scientific and technical works focus on stabilizing the metrological properties (measurement error, stability of the error over time, variation of the error with respect to influence quantities, e.g., environmental factors, and vehicle speed) of WIM systems. This will minimize the risk of incorrect classification of a normative vehicle as overloaded. The reason for such an error may be the influence of various internal or external factors on weighing accuracy and, as a result, the change of this accuracy even over short periods of time (i.e., hours).

As can be seen from the literature review presented above, the development of WIM systems capable of weighing vehicles with the highest possible and constant accuracy is the subject of legislative and technical activities carried out in many countries. Our paper also deals with this issue. Previous studies have shown that the real factors influencing the accuracy of weighing are, above all, the temperature of the road surface and the speed of the weighed vehicle. The weighing error caused by the daily temperature change may be as high as several dozen percent, depending on the technology of the load sensors used. Such a strong impact is a very serious limitation for the use of WIM systems in the direct mass enforcement mode. The intensity of influencing factors cannot be stabilized or controlled, but it can be measured. In the paper, we proposed a method of continuous monitoring of the uncertainty of weighing results caused by this influence. The basis of this method are the results of measuring the intensity of the indicated influencing factors. The use of this method will make it possible to solve the above-mentioned problem, i.e., it will make the weighing result credible. The validation of the result consists in ensuring that its uncertainty does not exceed the assumed level. At the same time, the proposed method allows to detect those cases in which the excessive intensity of influencing factors causes the uncertainty of the weighing result to exceed the assumed level. In this case, the weighing results can only be used for traffic monitoring purposes. Their second application

is the pre-selection of overloaded vehicles and directing them to weighing on static or Low-Speed scales.

The work is organized as follows: The second part discusses the influence of various factors on the accuracy of weighing. This analysis is based on previous research by the authors, as well as the work of other research teams dealing with this issue. The third part presents a concept for accuracy maps of WIM systems. In the fourth part, we discuss a methodology for determining accuracy maps. In the fifth part, we present extensive experimental material from two WIM systems located in Poland, in Gardawice and Grodziec. The experiments carried out at these stations lasted several months. Over that time, the results of weighing almost 30,000 trucks at each station were collected. The work ends with a summary presented in part six.

## 2. Influence Factors Limiting the Accuracy of WIM Systems

In WIM systems, axle load sensors are embedded in the pavement perpendicular to the traffic flow direction. Regardless of the sensor technology (quartz, bending plate, load cells), the fulcrum for the sensor is the pavement, which becomes a part of the weighing system. In this way, the properties of the pavement affect the properties of the whole WIM system [15].

Thus an important cause of weighing errors in WIM systems is their high sensitivity to the influence of environmental factors. There is a particularly strong influence of pavement temperature, but other factors include the influence of vehicle speed, driving technique (e.g., acceleration), wind force and direction. Previous studies have identified the impact of some of these factors on vehicle weighing errors. The tests were carried out for polymer sensors and quartz sensors. The intensity of this influence depends mainly on the axle load sensor technology [16,17] and on the properties of the pavement [15].

Many works focus on the influence of temperature on the weighing results of vehicles [17–20]. Long-term properties (e.g., sensitivity) of sensors resulting from the mechanical impact of vehicle wheels are also extremely important [16].

Changes in temperature affect the internal parameters of the sensors, as well as the mechanical parameters of the pavement [15]. A quantitative description of this impact, especially in relation to the pavement, requires long-term tests, carried out at the station, after the installation of the WIM sensors. The results of research on load sensors carried out in a climatic chamber were published in our work [21]. This paper presents the results of the study of the influence of temperature on the accuracy of weighing vehicles in WIM systems equipped with piezoelectric load sensors. The tests were carried out both in the climatic chamber and on the WIM road site. The obtained results indicate that in such a WIM system, a change in the surface temperature in the range of −10 °C–+30 °C causes a weighing error of up to 30%.

The combined influence of temperature changes in the properties of the sensor and the pavement in which it is installed results in a significant reduction in weighing accuracy. The results of research on load sensors installed in a bituminous pavement were published in the works [21,22].

In addition, vehicle speed also affects the accuracy of weighing. This observed effect has two sources. Firstly, the increase in speed translates directly into the enhancement of the dynamic component in the signal of the load exerted by the vehicle wheels on the ground. The second source is the deformation of the pavement under the influence of wheel loads. The longer this load lasts, the greater the deformation and the stronger the sensor response. As the susceptibility of the pavement depends, among other things, on its temperature, as result the sensor response is sensitive to both the pavement temperature and the speed of the vehicle [15]. The impact of both is correlated. The results of research carried out by the authors in this area are presented in the work [23], and the results of similar studies can be found in [17].

The results of tests of the impact of wind on the vehicle weighing result in a WIM system are presented in the paper [23]. The research was carried out at a station equipped

with quartz load sensors. However, due to the mechanism of this phenomenon (transfer of the load between axles or wheels), it should be expected that a similar effect will also be observed in the case of other load sensors. An observed change in wind direction can cause a change in the weighing result of up to 2%.

The results of the research published in the above-mentioned works allow for the formulation of the following conclusions:

- There is a correlation between the influence of temperature and vehicle speed on the weighing error;
- In the case of quartz and bending plate sensors, the temperature sensitivity is in the range of 5–10% depending on the type of pavement, and for polymer sensors it is much higher, reaching up to 50%;
- Change in the speed of the weighed vehicle in the range of 50–90 km/h causes an approximately 10% change in the weighing result in the case of polymer sensors and approximately up to 4% change in the case of quartz and plate sensors;
- Change in the wind direction can cause a vehicle weighing error of up to 2%, regardless of the type of load sensors used;
- These factors, i.e., surface temperature, vehicle speed, and wind direction, are the basic factors influencing the accuracy of weighing in WIM systems. However, the influence of other factors, such as precipitation, snow, and icing of the surface or humidity, cannot be neglected. A quantitative description of these influences requires further research, conducted at a special WIM station.

## 3. Accuracy Maps Concept

Due to the role that enforcement WIM systems should play, certain specific requirements are formulated. The most important include:

- Minimization of weighing error, which directly affects the efficiency of eliminating overloaded vehicles from traffic;
- Minimization of the likelihood of misclassification of the vehicle as overloaded;
- Equipping of administrative WIM systems with sensors that measure the intensity of factors affecting weighing results.

The accuracy of weighing in WIM affects the effectiveness of eliminating overloaded vehicles from traffic. This is due to the need to increase the permissible value of the axle load or the gross vehicle weight by the value of the weighing error. Such a procedure is aimed at avoiding the erroneous classification of a normative vehicle as an overloaded one. The paper [1] attempted to assess the minimum accuracy of vehicle weighing necessary to ensure sufficient efficiency in the elimination of overloaded vehicles from traffic. It has been established that an error in determining the gross weight of the vehicle not exceeding 2% ensures an acceptable level of efficiency for this process.

In weighing vehicles, two kinds of errors can be made. The error of the first kind is that a vehicle that is actually overloaded is not eliminated from traffic. This limits the efficiency of this process. An error of the second kind causes a standard vehicle to be considered overloaded. It seems that from a social point of view, the error of the second kind is more critical, as it results in administrative action against the carrier or shipper. This would be contrary to the social sense of justice.

Minimizing the likelihood of the second kind of error is only possible by ensuring the stability of weighing accuracy in WIM systems. Due to the influence of many external factors on accuracy, ensuring the stability of the system, especially over a longer period of system life, is not an easy task. The concept of accuracy maps of WIM systems proposed by us in this work is aimed at protecting against the negative effects of variability of weighing accuracy. Thanks to this, the proposed solution minimizes the likelihood of the occurrence of the second kind of error.

By the term accuracy map of the WIM system, we mean the area determined in the multidimensional space of factors affecting the results of vehicle weighing (temperature, speed, wind, precipitation, icing, etc.). Within this area, the weighing error shall not exceed

the value accepted as permissible for the weighing of vehicles. In other words, the accuracy map determines the acceptable intervals of the variability of influential factors for which the vehicle weighing error is acceptable. Figure 1 shows an example of a WIM accuracy map. For simplicity, this is a two-dimensional map, taking into account the influence of vehicle speed and pavement temperature on the weighing error.

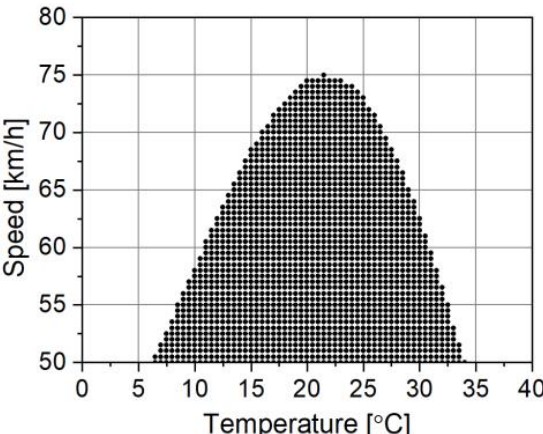

**Figure 1.** Example of the two-dimensional accuracy map of a WIM system, determined for the assumed limit value of relative weighing error.

The Figure 1 should be interpreted as follows: the variability of temperature and speed of vehicles inside the black area does not significantly affect the accuracy of weighing, and the error remains at an acceptable level. However, if the weighing takes place at a temperature or speed value outside the range determined by the boundaries of the accuracy map, this result should be rejected because the weighing error outside the blue area is too large for enforcement purposes.

An accuracy map constructed in this way allows for the verification of the reliability of the weighing result of each vehicle. The boundaries of the accuracy map determine the intervals of permissible variability of factors affecting the accuracy of weighing. Verification of the reliability/correctness of the weighing result is carried out by comparing the boundaries of the accuracy map with the result of measuring these influencing factors. If the value of any influencing factor exceeds the value resulting from the boundaries of the accuracy map at the time of weighing the vehicle, such a result should be considered unreliable. In this case, the probability of an error of the second kind and the misclassification of the normative vehicle as overloaded decreases.

The presented concept results in the third requirement formulated above for WIM systems for direct enforcement, namely the need to equip such systems with sensors to measure influencing factors. The research conducted indicates that the significant factors influencing the accuracy of weighing in WIM systems are the temperature of the surface, the speed of the weighed vehicle, and the direction and strength of the wind.

The measurement of the speed of vehicles weighed in WIM systems is not a problem. More complex is the problem of measuring the temperature of the surface. When planning the location of the WIM system and the placement of temperature sensors, it is necessary to take into account the recommendations which we have formulated. First of all, temperature sensors should be placed in close proximity to each axle load sensor and installed at the same depth. Additionally, the WIM station should not be located in a place where there are tall buildings or trees. Such objects can cause shading of the WIM station and a gradient of the pavement temperature. The longer the WIM site, the greater the probability of such an adverse phenomenon occurring, e.g., in the case of MS-WIM (multisensors—WIM). Moreover, the placement of the WIM station in a location surrounded by high objects is not advantageous due to the impact of the wind on the weighed vehicle. Such objects may cause different aerodynamic conditions to prevail in different parts of the WIM system.

## 4. Methodology for Determining Accuracy Maps

The basis for determining the accuracy map are multidimensional characteristics describing the dependence of the weighing error on the intensity of influencing factors. To determine such characteristics, the method of characteristic vehicles described in our work [23] can be used.

The characteristic vehicle method is carried out analogously to the pre-weighed vehicle method used during the calibration of WIM systems. In the pre-weighed method, the weighing result of the vehicle obtained at the WIM station is compared with the weighing result of the same vehicle on an accurate static scale. On this basis, the weighing error is determined, for example, as the relative difference between the two results, related to the static weighing result.

In the characteristic vehicle method, the pre-weighed vehicle is replaced by characteristic vehicles, i.e., vehicles with certain specific characteristics. It has been shown that the load of the first axle of five-axle vehicles (2-axle tractor unit + 3-axle semi-trailer) is distinguished by the smallest random variability and the smallest correlation with the gross weight of the vehicle among all classes of trucks. Therefore, the first axle load of this class of vehicles can be used as a reference in the assessment of the weighing error at the WIM station. At the same time, vehicles of this class have special design features (number of axles, distance between axles) that allow easy detection of their passage through the WIM station.

In the characteristic vehicle method, the weighing result from the WIM system of the first axle of the characteristic vehicles is compared with the mean value of the first axle load evaluated based on weighing results from accurate static scales. Such accurate vehicle weighing is carried out by vehicle inspection services (in Poland by inspection of road transport—IRT). To calculate the mean value (reference value) of the first axle load of the characteristic vehicles, we analyzed over ten thousand weighing results of the characteristic vehicles from the IRT.

The weighing error is determined by comparing the weighing results of the characteristic vehicles at the WIM station with the reference value. For the influence of disturbing factors on weighing accuracy to be visible, the tests must be carried out over a long period of time (6–12 months) while simultaneously measuring the intensity of these factors. A single point of such a characteristic is determined as the arithmetic mean of many values of the relative weighing error, obtained under specific conditions, i.e., for example, a temperature lying within the selected range with a breadth of 1 °C and a speed lying within the range with a breadth of 1 km/h. By repeating such averaging for subsequent, fixed values of temperature and speed of the weighed vehicle, we will obtain a set of average values of the weighing error for different values of temperature and speed, contained in the two-dimensional variability area of both influencing factors. In this example, the results obtained are presented in a three-dimensional coordinate system. On the axes of this figure there are respectively: temperature value, speed and the average weighing error occurring for these temperature and speed values. In this way, multidimensional weighing error characteristic is determined depending on influencing factors.

The accuracy map could be determined as a result of the intersection of the multidimensional error characteristic with the surface perpendicular to the error axis, at the level resulting from the adopted/required accuracy of the WIM system.

However, the random component of the weighing error means that the mean value of the error determined at each point of this characteristic contains a significant random component. For this reason, an accuracy map determined directly on the basis of the error characteristic would have very irregular boundaries, which would hinder its practical use. Therefore, we propose approximating the error characteristics of the surface.

In order to better present the concept of accuracy maps, we will use a graphic illustration. For this purpose, synthetic data was used. A broader discussion based on the results of experimental research will be presented in the next chapter.

For obvious reasons, we have limited this illustration to a three-dimensional case. Let us assume arbitrarily that the three-dimensional characteristic of the error has the shape shown in Figure 2a, while its approximation by the surface of the third degree is shown in Figure 2b.

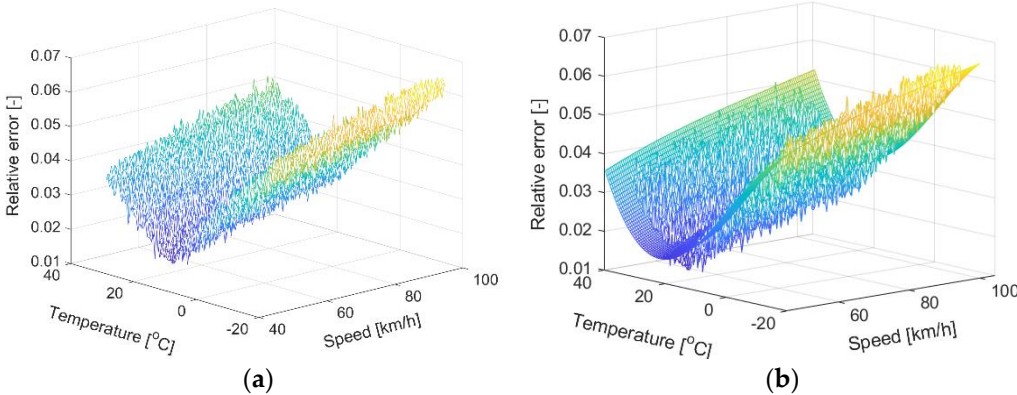

**Figure 2.** (**a**) Three-dimensional characteristic of the WIM system relative error; (**b**) the result of its approximation by the third-degree surface.

By intersecting the approximating surface, e.g., at the level corresponding to the value of error 0.025, we will obtain the map of accuracy presented in Figure 1.

It should be emphasized that the presented idea of determining the accuracy map must be repeated separately for each WIM station. WIM stations may differ in the quality and type of road surface and the technology of the load sensors used, and perhaps also in the way they are installed. For this reason, the ability to transfer accuracy maps between different WIM stations is limited. In each such case, it is required to analyze the impact of the above-mentioned factors on the weighing error or to designate a new accuracy map, dedicated to this location of the WIM station. This is a time-consuming process that requires many experiments. On the other hand, the use of the accuracy map significantly increases the reliability of weighing results.

Accuracy map building is performed off-line, so the computational complexity of this procedure is not significant. The practical use of the already built accuracy map, however, is computationally simple and consists only in checking whether the current values of the intensity of influencing factors are within the acceptable limits resulting from this map.

## 5. Results of Experimental Research

Experimental research was carried out at two WIM stations installed on the DK81 national road in Gardawice and on the DK46 road in Grodziec, southern Poland. The station in Gardawice is equipped with 16 lines of piezoelectric sensors, and the station in Grodziec with two lines of quartz sensors. Although polymer sensors, due to their limited accuracy and stability, are not used in WIM systems operating in direct mass enforcement, the possibility of comparing the results of their tests with sensors made in another, better technology, allows for a good illustration of the practical usefulness of accuracy maps. The load sensors, 4 m long each, were installed across the traffic lane, perpendicular to the road axis, at a distance of 1 m from each other. In addition, Pt100 temperature sensors were installed in the road surface, in the immediate vicinity of each load sensor. The installation depth of the temperature sensors corresponded to the depth at which the sensitive element of the load sensor was located. Such arrangement of the load and temperature sensors allowed for multiple measurements of the dynamic load of each axle of the passing vehicle and simultaneous measurement of the surface temperature in the place where the load sensor was installed. In addition, the knowledge of the distance between successive load sensors allowed to determine the speed of the vehicle passing through the WIM site. The results of the load measurement of each axle obtained from individual sensors were averaged (arithmetic mean). The number of averaged results corresponded to the

number of installed load sensors and was equally appropriate: 16 sensors in Gardawice and 2 sensors in Grodziec. The determined average value is an estimate of the static load of each axle of the weighed vehicle. The estimated gross vehicle weight (GVW) of the vehicle was determined as the sum of the estimates of the static loads of its axles. The site in Grodziec was additionally equipped with an anemometer to measure the direction and speed of the wind.

Measurement data from both WIM sites were collected at different times and contain a different number of weighing results. A total of four datasets were collected. The basic parameters of the collected data sets are summarized in Table 1.

**Table 1.** Characteristics of the sets of weighing results from two WIM sites.

| No. | Data Set Name | Data Collection Period | | Number of Weighing Results |
|---|---|---|---|---|
| | | **From** | **To** | |
| 1. | Gardawice | November 2005 | June 2006 | 28,673 |
| 2. | Grodziec 1 | August 2018 | November 2018 | 17,443 |
| 3. | Grodziec 2 | June 2018 | September 2018 | 12,796 |
| 4. | Grodziec 3 | March 2018 | April 2018 | 4184 |

An important problem in determining the accuracy maps of WIM systems is the need to collect a large number of weighing results. Weighing should be carried out under known conditions determined by the intensity of the influencing factors prevailing at the time of weighing. At the same time, they should cover the expected range of variability of these factors observed over a longer period of time in the location of the WIM system. This problem is illustrated by the characteristics shown in Figures 3 and 4 for different WIM sites Gardawice and Grodziec1 respectively. Characteristics were determined for the discretization of temperature with a step of 1 °C and speed of 1 km/h.

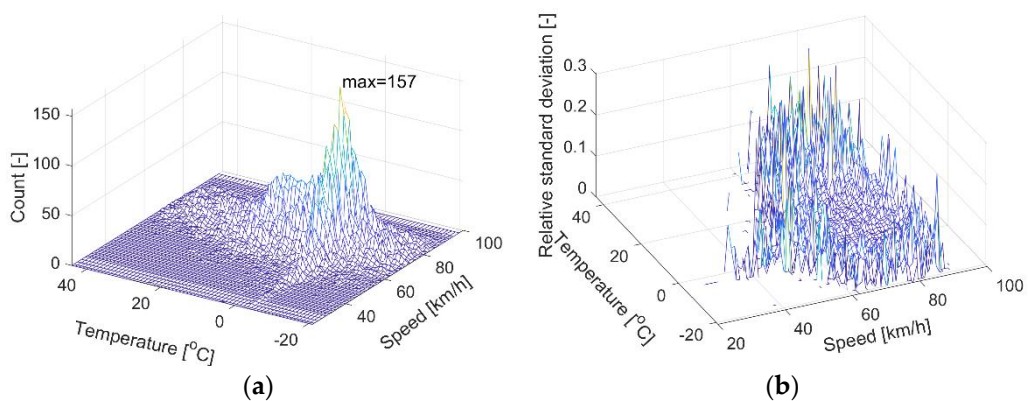

(a)	(b)

**Figure 3.** Gardawice: (**a**) the number of weighing results; (**b**) the relative standard deviation of the averaged weighing results in each temperature and speed range.

Due to the insufficiently long period of collection of measurement data, the number of weighing results in the range of extreme temperature and speed values is not sufficient. Thus, the relative standard deviation of the weighing error, averaged over the two-dimensional ranges of the influence factor variability (with a resolution of 1 °C and 1 km/h), increases at the boundaries of accuracy map. A much longer experiment would be required to ensure a comparable level in the determination of all WIM system error characterization points, within the full range of expected changes in influence factors.

On the basis of each data set, the three-dimensional characteristics of weighing errors, its approximation by the surface of both the second and third-degree and two-dimensional accuracy maps were determined.

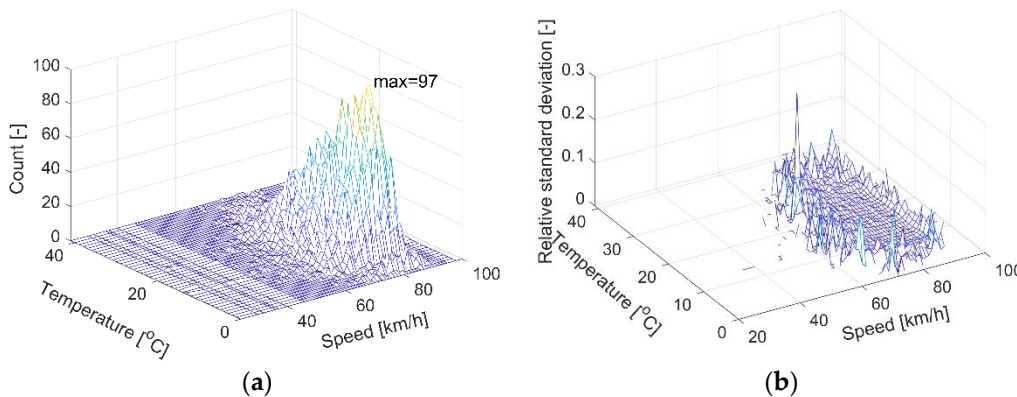

**Figure 4.** Grodziec1: (**a**) the number of weighing results; (**b**) the relative standard deviation of the averaged weighing results in each temperature and speed range.

Figures 5–7 show the three-dimensional characteristics of errors and 3rd-degree approximation surface determined on the basis of the collected measurement data.

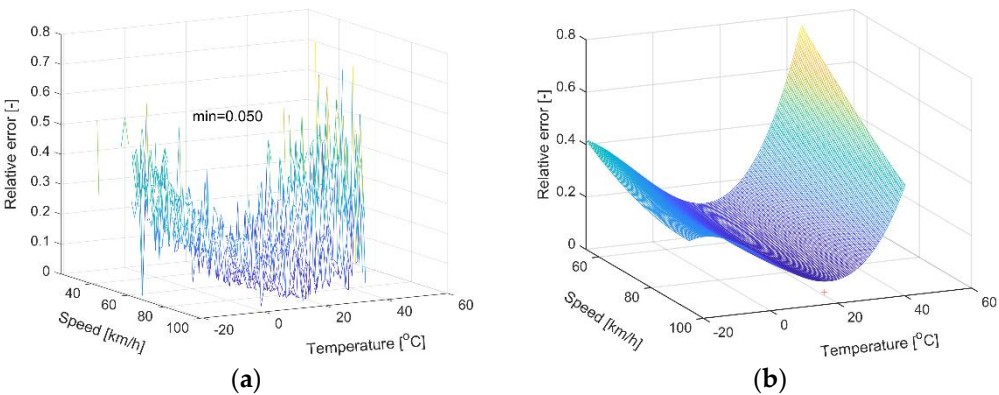

**Figure 5.** (**a**) Averaged relative error of measurement results from Gardawice (piezoelectric sensors); (**b**) third-degree approximating surface. The minimum error value on this surface, 0.050, was reached for a speed of 93 km/h at a temperature of 21.6 °C.

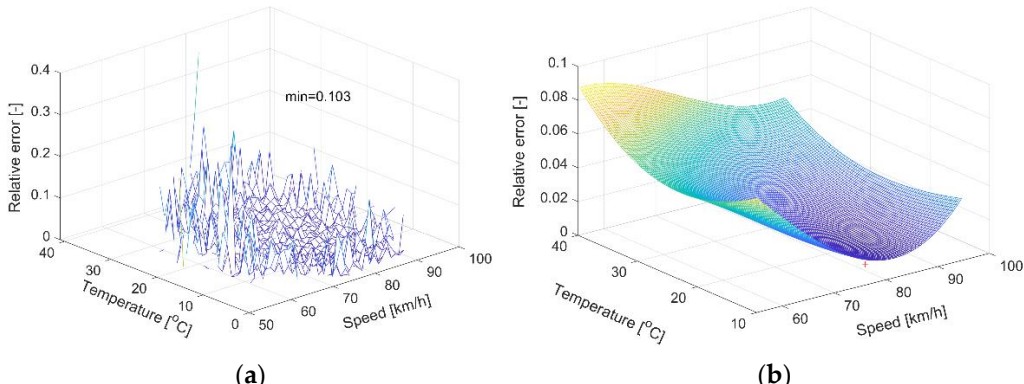

**Figure 6.** (**a**) Averaged relative error of measurement results from Grodziec1 (quartz sensors); (**b**) third-degree approximating surface. The minimum error value on this surface, 0.0103, was reached for a speed of 81.8 km/h at a temperature of 15.1 °C.

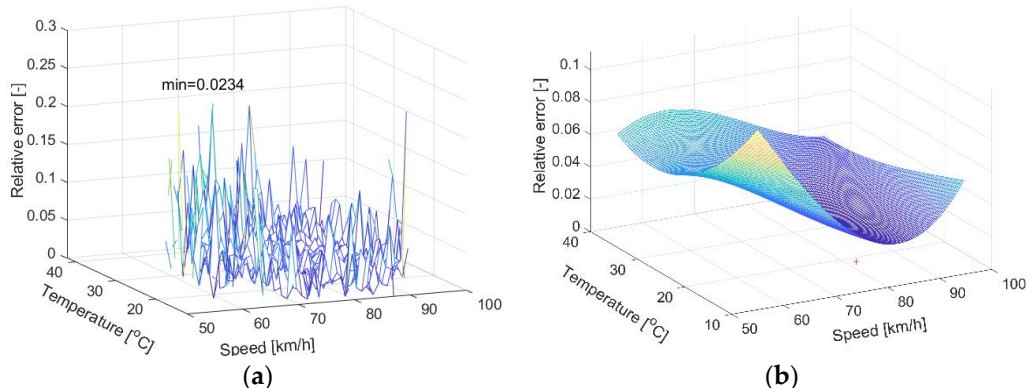

**Figure 7.** (**a**) Averaged relative error of measurement results from Grodziec2 (quartz sensors); (**b**) third-degree approximating surface. The minimum error value on this surface, 0.0234, was reached for a speed of 81.8 km/h at a temperature of 18.7 °C.

A comparison of the characteristics shown in Figures 5–7 leads to the following conclusions:

- There is clearly a greater influence of temperature on the weighing error of vehicles at the WIM station in Gardawice (a station equipped with polymer load sensors) compared to the stations in Grodziec, equipped with quartz sensors;
- The lower temperature sensitivity of the quartz sensors allowed for clearer visibility of the impact of the speed, which is similar for the sensors manufactured using both technologies;
- The minimum value of weighing error for quartz sensors is 1–2%, while for polymer sensors this is as much as 5%.

The last conclusion is confirmed by the two-dimensional accuracy maps shown in Figure 8 evaluated for data from the Gardawice site. The figure illustrates the boundaries of variation of both influencing factors (temperature and speed) for the assumed limit of relative weighing error, respectively 0.1 and 0.2. The accuracy maps were determined on the basis of both the second and third-degree approximating surface.

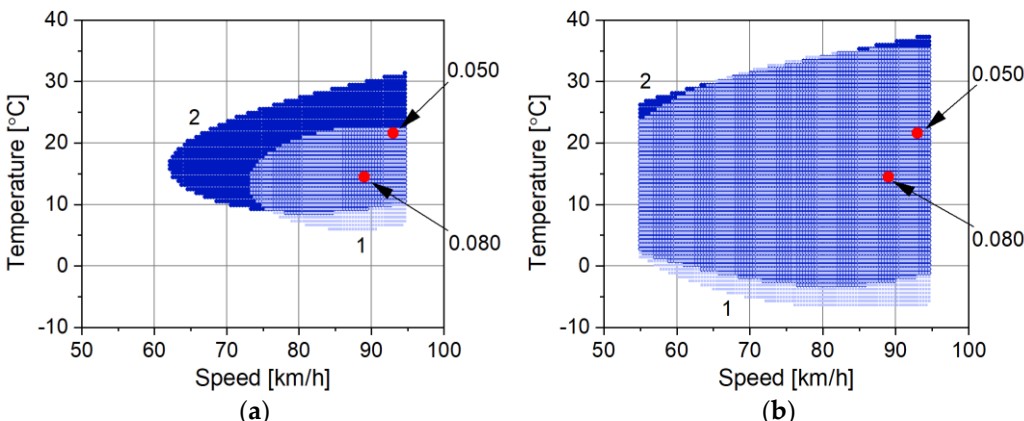

**Figure 8.** Comparison of 2D accuracy maps for Gardawice data, for (**a**) 0.1 and (**b**) 0.2 limit error and approximation of measurement data by the surface of the second (1) and third (2) degree. The minimum error value, 0.050, was achieved for a speed of 93 km/h at a temperature of 21.6 °C, using approximation with the surface of the third degree.

Reducing the degree of the approximation surface (Figure 8) causes the three-dimensional characteristic of the weighing error to become more strongly averaged, which results in narrowing the boundaries of permissible variability of influence factors. The accuracy map determined on this basis is "safer" for administrative applications, i.e., it minimizes the probability of misclassification of a normative vehicle as an overloaded vehicle.

Figures 9 and 10 illustrate accuracy maps for the WIM site in Grodziec.

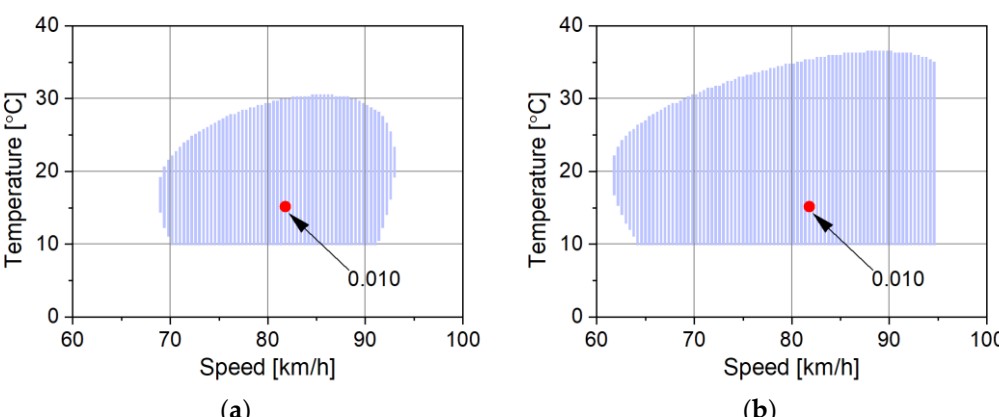

(**a**)    (**b**)

**Figure 9.** 2D accuracy maps for data from Grodziec1, determined on the basis of the approximation surface of the third degree, for the limit error of (**a**) 2.5% and (**b**) 4.0%. The minimum error value, 0.0103, was reached for a speed of 84.2 km/h at a temperature 13.9 °C.

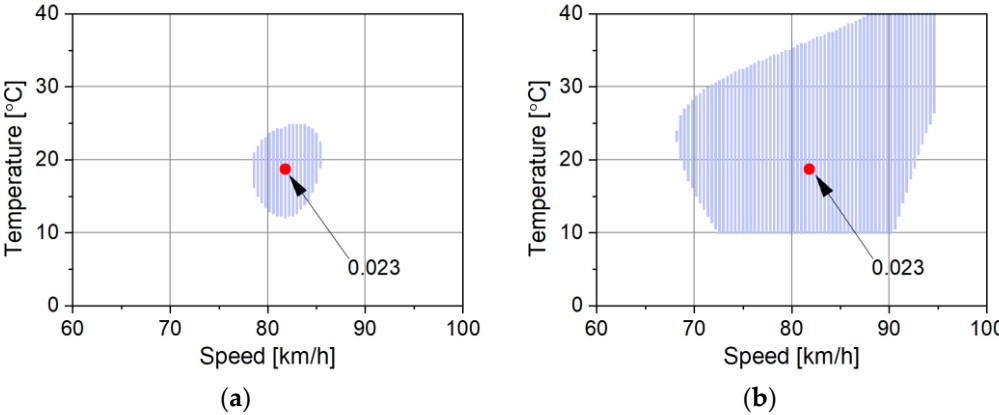

(**a**)    (**b**)

**Figure 10.** 2D accuracy maps for data from Grodziec2, determined on the basis of the approximation surface of the third degree, for the limit error of (**a**) 2.5% and (**b**) 4.0%. The minimum error value, 0.0234, was reached for a speed of 81.9 km/h at a temperature 18.7 °C.

The comparison of the accuracy maps determined for the WIM station equipped with polymer load sensors (Figure 8) and quartz sensors (Figures 9 and 10) shows a very large difference in the accuracy of weighing at these stations. The more accurate of the two is the WIM system equipped with quartz sensors.

From Figures 9 and 10 it is clear that the accuracy maps for different time periods for the same system differ from each other. For the August-November period (Grodziec1 dataset), the accuracy of vehicle weighing is higher than for the June-September period (Grodziec2 dataset). The minimum value of the weighing error is lower and the boundaries of permissible variability of influence factors are wider.

In the context of the proposed methodology for determining accuracy maps for WIM systems, it is possible to formulate a question about the role of approximating surfaces. The weighing results contain a significant random component, hence even after their averaging, the error values determined for the chosen operating conditions of the WIM system also contain a random component. The boundaries of the accuracy map determined directly on the basis of measurement data would be very irregular, which would significantly hinder its practical use. The approximation of experimental results with a second or third-degree surface smooths these results, and the accuracy map is thus more regular. This is illustrated by the accuracy maps shown in Figure 11.

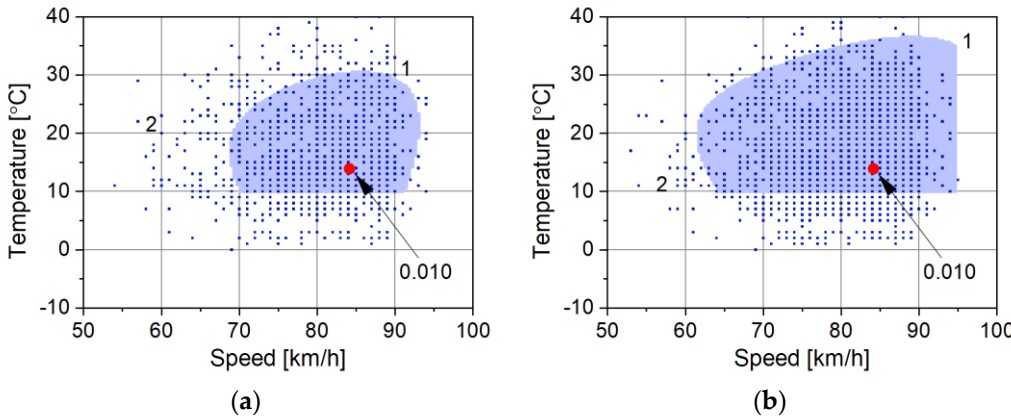

**Figure 11.** Accuracy maps determined both on the basis of the surface approximating the experimental results (1) and directly on the basis of the three-dimensional error characteristic (2), for the limit error of (**a**) 2.5% and (**b**) 4.0%.

Accuracy maps determined for the same WIM system in different periods of its operation allow for an assessment of the non-stationarity of metrological parameters of the system. Such maps for three periods of operation of the WIM system installed in Grodziec are shown in Figure 12.

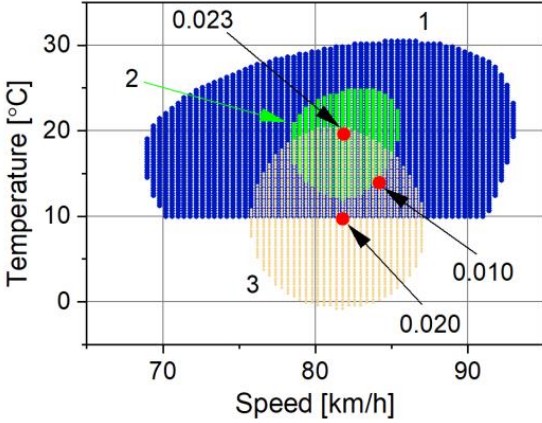

**Figure 12.** Comparison of accuracy maps for the Grodziec system, determined from different periods of operation of the WIM system. 1—Grodziec1; 2—Grodziec2; 3—Grodziec3. Approximation of third-degree surface error value: 2.5%.

A comparison of these maps shows that this system had the highest accuracy in the Grodziec1 period of operation. On the accuracy map, this is revealed by the smallest weighing error value and the broadest boundaries of permissible changes in influence factors that do not yet exceed the assumed limit error value. It can be assumed that the length of the time interval since the last calibration of the WIM system has a significant impact.

## 6. Summary

The main achievement of the presented paper is the method for determining and applying accuracy maps for WIM systems. It is a tool that allows to verify the reliability of the vehicle weighing result. Thus, it is possible to reduce the probability of making a mistake consisting in indicating a standard vehicle as an overloaded one. This helps to avoid unjustified imposition of administrative penalties. As the presented literature review shows, efforts are underway in many countries for the practical implementation of WIM systems operating in the direct mass enforcement mode. This is an important task from the point of view of effective protection of road infrastructure, ensuring the safety of road users

and protecting the natural environment. According to the authors, the use of the proposed accuracy maps brings closer the achievement of this goal.

The reason for weighing error is the significant sensitivity of WIM systems to environmental factors as well as to the speed of the weighed vehicle. The experimental results from the two WIM systems, equipped with load sensors made with different technologies, allow us to illustrate the proposed method of determining the accuracy maps and their use in order to confirm the reliability of the weighing result of each vehicle. A significant limitation of this method is the need to collect a very large number of weighing results at a specific WIM station, covering a wide range of changes of influence factors. This problem can be effectively solved using the characteristic vehicle method. Therefore, long-term observation of the WIM system is necessary. The current research into the concept of accuracy maps is a pilot study. At this stage of the research, an accuracy map created for a specific WIM system can be used to validate measurements from that particular system. Due to the multitude of factors influencing measurement accuracy, further research is needed to determine how the accuracy maps could be generalized and which factors influencing measurement could be transferred to other WIM stations of a given type. Particularly relevant here could be research related to the influence of surface type and temperature on measurement accuracy with the aim of creating generalized accuracy maps. However, in general the proposed method can be successfully used to assess the quality of operation of other technical facilities subject to the influence of disturbing factors.

**Author Contributions:** Conceptualization, J.G.; methodology, J.G.; software, J.G. and P.B.; validation, P.B., R.S. and M.D.; formal analysis, J.G. and R.S.; investigation, J.G., P.B., R.S. and M.D.; writing—original draft preparation, J.G.; writing—review and editing, R.S. and M.D.; visualization, P.B.; supervision, J.G.; project administration, R.S.; funding acquisition, J.G. and R.S. All authors have read and agreed to the published version of the manuscript.

**Funding:** This research was funded by Polish Ministry of Education and Science, in the program "Polish Metrology", grant number: "PM/SP/0041/2021/1".

**Data Availability Statement:** The data presented in this study are available on request from the corresponding author. The data are not publicly available up to time of finishing the project.

**Conflicts of Interest:** The authors declare no conflict of interest.

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
