# Peer review of "Accuracy Maps of Weigh-In-Motion Systems for Direct Enforcement"

_electronics, doi:10.3390/electronics12071621_

Round 1
Reviewer 1 Report
It is seen that authors provide a combined citation as [1-13] in the Introduction section but it does not provide pretty much information about the legislative and technical topics. It is required to present detail information on these references to create a useful motivation on readers’ end. Otherwise, this citation method does not create an effect.
The literature survey, motivation and contributions of the proposed study were not clearly addressed in the Introduction. The importance of the conducted study should be presented in detail.
Authors cite their own paper [20] in the 2nd section and advice reader to see the paper for details. However, the related part of that paper to this study should be described in the required part with sufficient information, and readers are get informed about the connection of both studies.
The blue area described for Fig. 1 on line 156 on pg. 4 is missing on Fig.1. Please check and revise the description.
The methodology of the proposal was not clearly addressed and not described in the paper. Authors cite to other studies until 5th section and two figures are presented. The overall organization of the paper does not provide sufficient understanding on the proposed method or study.
Detailed information is required for studies on how the sensors are located, which parameters and approaches were used to determine exact locations, how the data was inherited and processed etc.
There is not any evidence presented on the accuracy of the presented analysis results in the figures under Experimental study section. Therefore overall organization of the paper should be extensively improved.
Conclusion is also lacking in addressing the main contribution of the proposed study. It should be improved.
Author Response
The corrected text is shown in red, while the deleted fragments are crossed out and marked in blue in the text of the manuscript. The explanatory file is attached.

Reviewer 2 Report
This article describes experimental results with WIM sensors under different conditions. As result there are maps associated with statistical results of two WIMS on road in Poland. To this point it brings some engineering and itneresting results. Further there is a desrciption of method how to get accuracy map but not how use results for generally purposes. Using is very complicated still. There is a need of longer experiment. There is no description of data processing method.
In the end, there is a great work done but I think there is missing information about data processing, time consumption of operatins and method of realization in general with portability for different locations and lands.
There is big number of self citations and some references are not visible in scopus or different databases to check these sources.
Author Response

(The authors gave the same response as above.)

Reviewer 3 Report
The authors presented a concept of mapping the multiple environmental parameters corresponding to the measurement error below the acceptable value for the weighing of vehicles. In my opinion, the article can be accepted for Electronics.
Several comments, which I suppose could improve the article:
—Using both “error” and “uncertainty” terms in the text could be confusing for the reader
—The citation in line 32 could be useful.
—That is the meaning of “ metrological properties” terms?
—Is there a physical reason to use a 3rd-degree approximation for the three-dimensional characteristic of errors?
—Figures 3b and 4b are difficult to understand. I propose to present data as sets of 2d curves, cuts.
— An example of the measuring system and more details about measured data elaboration can be useful.
Author Response

(The authors gave the same response as above.)

Round 2
Reviewer 1 Report
The proofreading should be performed for the submission. My comments are responded, and accepted as sufficient.
Author Response
The paper was verified by a native speaker and corrections were made in the text in “Track Changes” function.
Reviewer 2 Report
Thank you for paper improving. I found next questions about using this measurement results but this article is about maps creation process.
Author Response
The paper was verified by a native speaker and corrections were made in the text in “Track Changes” function.
Answer to reviewer was upload and text of the paper is suplemented.
